# Recurrent Water Deficit and Epigenetic Memory in Medicago Sativa L. Varieties

**Yannis E. Ventouris** [1,†]**, Eleni Tani**[1,*,†]**, Evangelia V. Avramidou** [2]**, Eleni M. Abraham** [3]**, Styliani N. Chorianopoulou** [4]**, Dimitrios N. Vlachostergios** [5]**, Georgios Papadopoulos** [1] **and Aliki Kapazoglou** [6]

[1] Department of Crop Science, Laboratory of Plant Breeding and Biometry, Agricultural University of Athens, Iera Odos 75, 11855 Athens, Greece, yannisventouris@gmail.com (Y.V.); etani@aua.gr (E.T.); gpapadop@aua.gr (G.P.)

[2] Laboratory of Forest Genetics and Biotechnology, Institute of Mediterranean Forest Ecosystems, Athens, HAO "DEMETER" Terma Alkmanos, Ilisia, 11528 Athens, Greece; email: aevaggelia@yahoo.com (E.V.A.)

[3] Laboratory of Range Science, School of Agriculture, Forestry and Natural Environment, Aristotle University of Thessaloniki, 54124 Thessaloniki, Greece, email: eabraham@for.auth.gr (E.A.)

[4] Department of Crop Science, Laboratory of Plant Physiology and Morphology, Agricultural University of Athens, Iera Odos 75, 11855 Athens, Greece, s.chorianopoulou@aua.gr (S.C.)

[5] Institute of Industrial and Forage Crops, HAO-DEMETER, 41335 Larissa, Greece; vlachostergios@gmail.com

[6] Institute of Olive Tree, Subtropical Crops and Viticulture (IOSV), Department of *Vitis*, Hellenic Agricultural Organization-Demeter (HAO-Demeter), Sofokli Venizelou 1, Lykovrysi, 14123 Athens, Greece; akapazoglou@gmail.com

**\*** Correspondence: etani@aua.gr; Tel.: +302105294625

**†** The authors contributed equally to the manuscript

**Abstract:** Global DNA methylation changes in response to recurrent drought stress were investigated in two common Greek *Medicago sativa* L. varieties (Lamia and Chaironia-Institute of Industrial and Forage Crops). The water deficit was implemented in two phases. At the end of the first phase, which lasted for 60 days, the plants were cut at the height of 5 cm and were watered regularly for two months before being subjected to the second drought stress, which lasted for two weeks. Finally, the following groups of plants were formed: CC (controls both in phase I and phase II), CD2 (Controls in phase I, experiencing drought in phase II), and D1D2 (were subjected to drought in both phase I and phase II). At the end of phase II, samples were taken for global DNA methylation analysis with the Methylation Sensitive Amplification Polymorphism (MSAP) method, and all plants were harvested in order to measure the fresh and dry weight of roots and shoots. The variety Lamia responded better, especially the D1D2 group, compared to Chaironia in terms of root and shoot dry weight. Additionally, the shoots of Lamia had a constant water status for CD2 and D1D2 group of plants. According to DNA methylation analysis by the MSAP method, Lamia had lower total DNA methylation percentage after the second drought episode (D1D2) as compared to the plants CD2 that had experienced only one drought episode. On the other hand, the total DNA methylation percentage of Chaironia was almost the same in plants grown under recurrent drought stress conditions compared to control plants. In conclusion, the decrease of DNA methylation of Lamia stressed plants probably indicates the existence of an epigenetic mechanism that may render drought tolerance.

**Keywords:** alfalfa; drought stress; stress memory; epigenetics; MSAP

---

## 1. Introduction

Climate changes affect crop production as well as the availability of natural resources, such as land and water. More than 60% of the annual yield fluctuations of important crops have been attributed to climatic variability [1]. Thus, the release of new, improved varieties combining high productivity and increased adaptability to the ongoing environmental changes has become an imperative [2].

Recent findings highlight a new source of heritable variation in plants, derived from epigenetic regulations, that play an essential role in directing the expression of agronomically important traits and adaptation of plants to a changing environment due to global climate change [3].

In this respect, since the epigenetic regulatory mechanisms are greatly affected by the environment, they could be a potential source of variation for adaptation [4]. Furthermore, as these mechanisms influence flowering regulation and reproductive development, they constitute an important factor in the plant breeding process [5]. Plant tolerance to external conditions depends on a complex network of molecular mechanisms, including massive changes in gene expression programs, regulated by epigenetic factors that modify chromatin structure. Chromatin remodeling to open (permissive) or closed (repressive) forms and subsequent gene activation or gene silencing, respectively, is mediated by DNA methylation, chromatin histone modifications, and the action of small RNAs [6–8].

Among the epigenetic regulatory mechanisms, DNA methylation is one of the most well-studied, especially in relation to plant adaptation to a broad range of abiotic stresses [9,10]. DNA methylation, the process whereby a methyl group is added to a cytosine or adenine nucleotide of DNA, plays a key role in the epigenetic regulation of eukaryotic gene expression [11–13]. Growing evidence indicates that the ability of plants to respond to environmentally adverse conditions such as drought stress is strongly dependent on changes in the DNA methylation status, which can be tissue- and cell type-specific, and can be propagated mitotically within a single generation. Importantly, these changes also can be heritable and stably transmitted to the offspring of the next generation [14].

A genome-wide study by [15] showed that drought-tolerant and drought-susceptible *Oryza sativa* varieties displayed differential site-specific DNA methylation upon drought imposed at the tillering stage, and nearly 70% of these DNA methylation changes were restored upon recovery under non-drought conditions [15]. In *Arabidopsis thaliana* [16] found that plants, which had been previously exposed multiple times to drought stress conditions, had the ability to respond to new stress by more rapid adaptive changes to gene expression patterns compared to plants which had not been exposed previously to drought conditions. Evidence already exists for *Arabidopsis thaliana* showing transgenerational effects associated with low humidity stress. These are transmitted to at least one generation by changes in the DNA methylation status at various sites of the *SPEECHLESS* (*SPCH*) and *FAMA* genes encoding basic-helix-loop-helix (bHLH) transcription factors [17,18]. Reports are emerging regarding other crops such as *Zea mays*. Forestan et al. [19] studied the response of maize plants to mild water stress, by modulating stress-induced chromatin mark changes that control memory genes in order to activate or suppress them in upcoming stress events.

*Medicago sativa* L. (*M. sativa*) or alfalfa is a perennial legume species that is extensively cultivated for grazing, hay, and silage production worldwide [20], and a key component of many crop rotation systems [21]. It is characterized by high forage production and high nutritional value for animal feed, with higher and lower content of protein and fibers, respectively, compared to other forage crops [22]. Generally, it is considered a drought-tolerant forage legume due to its deep rooting system [20] with wide within-species variation [23]. However, the anticipated increase in dry conditions (higher temperatures, less rainfall) in the future would make further improvement of this trait a top priority. The response of *M. sativa* to abiotic stresses has been widely studied at the physiological, genetic, and molecular levels. Nevertheless, there are limited studies on epigenetic regulation in relation to its response to abiotic stresses [24]. Regarding DNA methylation, only one study was conducted in *M. sativa* in order to investigate if salinity stress alters DNA methylation [25]. Findings suggested that

salinity increased global DNA methylation, particularly in plants exposed to the highest salinity levels [25].

The main scope of this study was to investigate global DNA methylation changes in response to recurrent drought stress in two varieties of *M. sativa* and to unravel possible epigenetic memory events that may be associated with better adaptation to repeated drought stresses.

## 2. Materials and Methods

### 2.1. Plant Material and Treatments

*M. sativa* variety Chaironia was developed after applying mass selection in an old American variety named African. It is an early maturing variety of good persistence even after the 4th year of cultivation, and can tolerate cold temperatures better than its maternal genetic material. It is considered to be moderately tolerant to drought, with a quick rate of regrowth after cuttings. It is better adapted in the hottest areas of Greece. The hay yield of Chaironia ranges between 0.11 and 0.14 t ha$^{-1}$ under irrigated fields [26]. Variety Lamia was developed after mass selection within a local Hellenic variety named Ypati 84. It belongs to a mid-early maturity group and has maintained high yield and quality even after the 4th year of cultivation. The rate of regrowth after cutting is very quick and is characterized by wide adaption capacity in various pedoclimatic conditions. The hay yield of Lamia ranges between 0.12 and 0.16 t ha$^{-1}$ under irrigated fields [26]. The seeds were sown in plastic pot trays containing a commercial growing medium. Five seeds were sown in each pot. The trays were then placed in a glasshouse for germination, and during that period, sufficient water was supplied so that the potting mix was constantly moist but not overwatered. One month after sowing, the seedlings were transplanted into 1 lt plastic pots containing a mix of peat moss and perlite, in the proportion of 4:1. The plants were thinned down to one seedling per pot and were subsequently subjected to two rounds of drought stress. The pots were placed completely randomly in the glasshouse and were rotated three times throughout the experiment.

The first round (phase I) started one month after transplanting and lasted for 60 days. During this period, 15 plants of each variety served as controls and were being watered with 70% of the substrate's field capacity, whereas 15 plants were subjected to drought stress by withholding water. The watering of control plants took place once every three days. Pots of control plants were weighted along with pots of stressed plants. When pots of stressed plants reached 30% of water pot capacity, the first round of drought stress was completed. The completion of this first phase of drought stress was marked by cutting the aboveground parts of all the plants at the height of 5 cm. After a period of two months, during which all plants were being thoroughly watered regularly and new stems from the crowns emerged, the second round of drought stress (phase II) initiated, which lasted for 13 days due to the fact that the plants were already grown and showed symptoms of drought stress very quickly. Of the former 15 control plants of each variety, eight were randomly chosen to experience water deficit conditions along with the 15 plants that had been already stressed during phase I. These were the treatments: CD2 (controls in phase I, experiencing drought in phase II) and D1D2 (drought-stressed both in phase I and phase II), respectively. Accordingly, the remaining seven plants of the initial controls served again as controls at phase II and are referred to as CC treatment plants (controls both in phase I and phase II). Watering of all plants took place every other day and the water availability conditions were optimal for the CC plants, whereas both CD2 and D1D2 plants were watered to 30% of pot capacity. This set of treatments (Figure 1) was specifically chosen to observe whether any epigenetic memory would occur as a result of the repetition of drought stress. On day 133, samples were taken for the methylation-sensitive amplification polymorphism (MSAP) analysis. Additionally, seven plants for each treatment for each variety were harvested in order to measure the fresh and dry weight of roots and shoots.

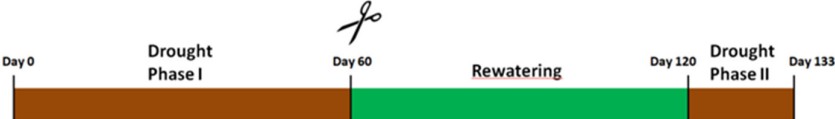

**Figure 1.** Schematic illustration of the experimental design. Two months after sowing, the first round of drought stress was carried out for 60 days. Then, the second period of water deficit was implemented.

### 2.2. Calculation of Substrate Pot Capacity

The volume of water retained by the growth medium during pot capacity (PWsat) was determined by weighing the pots before and after they were thoroughly watered. Subsequently, the mean weight of the pots before and after irrigation was calculated, and the difference between the two values revealed the mean weight of the maximum amount of water the substrate was able to hold. This process was performed twice, prior to each of the two drought periods. This measurement was used to determine the volume of irrigation water for all treatments as previously mentioned

### 2.3. Shoot and Root Dry Weight and Shoot and Root Water Content Determination

The fresh weight of both the aboveground and the root system was separately measured immediately after the plants had been removed from pots. The roots were thoroughly washed, and the excess of water was removed by filter paper. Dry matter was determined after drying at 70 °C for one week. The shoot and root water contents were determined as follows: first, the amount of water in each sample ($H_2O$) was estimated by subtracting the tissue's dry weight (DW) from its fresh weight. The ratio $H_2O$/DW was calculated as a measure of the samples' water content.

### 2.4. Sampling and DNA Extraction

One mature leaf as close as possible to the shoot apex was collected from three different plants in each entry (Lamia CC, CD2, D1D2, and Chaironia CC, CD2, D1D2). The leaves were immediately immersed in liquid nitrogen in order to avoid any potential changes in the DNA methylation status. Genomic DNA was extracted using the NucleoSpin Plant II kit (Macherey-Nagel, Duren, Germany). DNA quantification was determined using a Nanodrop spectrophotometer.

### 2.5. MSAP Procedure

MSAP or Methylation-sensitive amplified polymorphism is a modification of the standard amplified fragment-length polymorphism (AFLP) technique. *EcoRI* is being used as the rare cutter and the methylation-sensitive restriction enzymes *HpaII* and *MspI* as the frequent cutters due to the fact that the latter are a pair of isoschizomers, recognizing the same target sequence 5'- CCGG -3', but they have differential sensitivity to methylation at the inner or outer cytosine. For the MSAP procedure, from each sample, 200 ng of genomic DNA was digested with 4U of *EcoRI* and 3U of *HpaII*, and 200 ng was treated with 4 U of *EcoRI* and 3U of *MspI*. The digestion was carried out at 37 °C for 3 hours. The resulting DNA fragments and the *EcoRI* and *HpaII/MspI* adapters (Table 1) were ligated at 25 °C for three hours using 400 U/ul of T4 DNA ligase (New England Biolabs, Ipswich, Massachusetts, USA). After that period, the samples were subjected to heat shock treatment for 10 min at 65 °C to end the ligation reaction. A primer pair based on the sequences of the *EcoRI* and *HpaII/MspI* adapters (Table 1) with one additional selective nucleotide at the 3' end (*EcoRI*+A and *HpaII/MspI*+T) was used for the pre-selective PCR step. Pre-amplification PCR was performed in a total volume of 20 µl containing 1 × Kapa Taq Buffer, 0.4 u dNTPmix (10mM), 2.5 mM MgCl2, 30 ng of each primer *EcoRI*+A, *HpaII/MspI*+C, 1U Taq DNA polymerase (Kapa Biosystems, Wilmington, Massachusetts, USA) and 5 µl of diluted fragments (from the digestion and ligation reaction). The cycling program was the following: —Initially, a brief 30-sec hold at 94 °C was implemented,

followed by 23 cycles of 94 °C for 30 sec, 56 °C for 30 sec, and 72 °C for 1 min, and subsequently, followed by a final hold at 72 °C for 30 min. A 5 µl aliquot of the reaction was electrophoresed on agarose (1.25% w/v + ethidium bromide) to verify amplification; the remaining 15 µl were diluted 10-fold with TE. Selective amplifications were carried out in 10 µl total volumes containing 5 µl of a diluted pre-selective template and 0.2 dNTP mix (10 mM), 2.5 mM MgCl$_2$, 30 ng of *HpaII/MspI* primer, 30 ng of *EcoRI* primers, and 1U of Taq DNA polymerase (Kapa Biosystems, Wilmington, Massachusetts, USA) per reaction. Selective amplification cycling was performed according to the following program—an initial cycle of 94 °C for 30 sec, 65 °C for 30 sec, 72 °C for 1 min, then twelve cycles of 94 °C for 30 sec with an annealing temperature starting at 65 °C for 30 sec, but decreasing by 0.70 °C in each cycle, 72 °C for 1 min, and finally, 22 cycles of 94 °C for 30 sec, 56 °C for 30 sec, 72 °C for 1 min, with a final hold at 72 °C for 30 min. The pre-selective and selective primers used are given in Table 1

**Table 1.** Adapters and primers used for the MSAP analysis.

|  | 5′ to 3′ Sequence |
|---|---|
| *EcoRI* adapter | CTCGTAGACTGCGTACC |
|  | AATTGGTACGCAGTC |
| *HpaII/MspI* adapter | GACGATGAGTCTCGAT |
|  | CGATCGAGACTCAT |
| Pre-selective *EcoRI* primer | GACTGCGTACCAATTC-A |
| Pre-selective *HpaII/MspI* primer | ATGAGTCTCGATCGG-T |
| Selective *EcoRI* primers | GACTGCGTACCAATTC+ATG |
|  | GACTGCGTACCAATTC+ACT |
|  | GACTGCGTACCAATTC+AAC |
|  | GACTGCGTACCAATTC+AAG |
| Selective *HpaII/MspI* primer | ATGAGTCTCGATCGG+TCA |

*2.6. Data Collection and Statistical Analysis*

The dry weight of shoots, roots, their ratio, and the water content of shoots and roots were tested for normality using the Shapiro–Wilk test. All the parameters were normally distributed (p = 0.09, 0.10, 0.98, 0.48), respectively, except the water content of shoots and roots, which are log-transformed. The log-transformed water content of shoots was normally distributed (p = 0.18), while for roots was not. Additionally, the Levene test and the partial eta-squared test were applied in order to test the homogeneity of variances and the magnitudes of the main effects, respectively. Thus, the effect of treatment on the dry weight of shoot and roots was detected by two-way ANOVA. For not normally distributed data, the Kruskal–Wallis test was used. The Tukey test at 0.05 probability level was chosen for the detection of differences among means. The IBM SPSS Statistics 23 software (SPSS Inc., Chicago, IL, USA) was used for the statistical analysis.

An AFLP Excel Macro [27] was used to convert allele size data from GeneMapper4.0 (Applied Biosystems, Foster City, California, USA) into binary form and to indicate the presence or absence of fragments. Only reproducible fragments ranging from 150 to 500 bases were counted and further analyzed in order to reduce the impact of potential size homoplasy [28]. For MSAP analyses, a comparison of the banding patterns of *EcoRI/HpaII* and *EcoRI/MspI* reactions results in four conditions of a particular fragment; I: fragments present in both profiles (1/1), indicating an unmethylated state; II: fragments present only in *EcoRI/MspI* profiles (0/1), indicating hemi- or fully methylated CG-sites; III: fragments present only in *EcoRI/HpaII* profiles (1/0), indicating hemimethylated CHG-sites; and IV: the absence of fragments in both profiles (0/0), representing an uninformative state caused either by different types of methylation, or due to restriction site polymorphism [29]. The total analysis was performed in R software with MSAP_calc program and Mixed Scoring II method, which is a reliable program and method in order to access different methylation changes [29]. We further analyzed each category of methylated (h & m) and non-methylated (u) alleles with GenAlEx 6 [30] in order to compute the epigenetic Shannon Information

Index (Iepi) and haploid epigenetic diversity (hepi) within and between cultivars and treatments. The Kruskal–Wallis test was performed in order to examine significant differences between different treatments and cultivars.

Furthermore, we performed Dunn's post hoc tests with Bonferroni correction on each pair of groups for h, m alleles, and total methylation (h+m alleles) for Iepi. Due to the fact that the Shannon Diversity Index and genetic diversity are strongly correlated, and Shannon provides an alternative method of quantifying biological diversity across multiple scales [30], we further discussed results for Iepi. Moreover, the Shannon Information Index (I) is widely used in ecological studies as a measure of diversity, as it calculates allele differences [29,30] across loci of different samples.

Finally, we produced a heat map presentation, which is actually a permuted data matrix of the six different treatments in a three-variable classification space (h-, m-, and u-methylation). The distance measure is the Euclidean distance [31], and the joining method is single linkage, and the software used is SYSTAT12 (Systat Software 2007, San Jose, California, USA) [32]. The heat map presentation is actually a permuted data matrix where the Operational taxonomic units (OTUs) are reshuffled to form a matrix having a largest-smallest diagonal arrangement of distance values, which are visualized with the colors and their combinations in the legend.

## 3. Results

### 3.1. Shoot and Root Dry Weight

The drought treatments had a significant effect ($p < 0.05$) only on the dry weight of shoots and roots (Table 2). On the other hand, the varieties significantly affected ($p < 0.05$) the dry weight of shoots and roots, as well as the shoot/root ratio (Table 2). Moreover, the interaction between the drought treatments and varieties was statistically significant for shoot and dry root weights (Table 2), indicating that the drought treatments' effect on the varieties was not consistent. The Levene test was not significant ($p > 0.05$) in all cases indicating homogeneity of the variances. The eta-squared test was higher for the treatments (0.7) compared to the varieties (0.3). This means that the higher percentage of variance was explained by treatments compared to varieties.

The drought treatments significantly reduced both the shoot and dry root weight, while they did not affect the shoot/root ratio. The plants (D1D2) that underwent both phases of drought stress had the lowest shoot and root weights (Table 2). The shoot dry weight was significantly reduced under drought treatments for both varieties (Figure 2). However, the dry shoot weight of Lamia did not significantly differ between CD2 and D1D2, while that of Chaironia was significantly reduced (Figure 2). On the other hand, Lamia had significantly lower dry root weight only under D1D2 drought treatment, while the dry root weight of Chaironia was gradually reduced from the control to CD2 and D1D2 treatments (Figure 3). The shoot dry weight of the studied varieties did not significantly differ under control and CD2, but under D1D2, Lamia had higher dry shoot weight compared to Chaironia (Figure 2). Regarding the dry root weight, the studied varieties did not significantly differ under the control, but Lamia had a higher dry root weight under both CD2 and D1D2 drought treatments (Figure 3).

**Table 2.** Mean values +/- SE for each of the three different treatments (CC, CD2, and D1D2), as well as for each of the two varieties tested. Statistically significant differences between treatments (A), between varieties (B), and their interaction (AXB) are also presented.

| Treatment | Shoot Dry Weight (g) | Root Dry Weight (g) | Ratio: Shoot/ Root |
|---|---|---|---|
| CC | 6.95 ± 0.34a | 5.66 ± 0.30a | 1.28 ± 0.11a |
| CD2 | 4.04 ± 0.31b | 3.53 ± 0.27b | 1.25 ± 0.11a |
| D1D2 | 2.57 ± 0.35c | 1.79 ± 0.30c | 1.56 ± 0.12a |
| **Variety** | | | |
| Lamia | 5.10 ± 0.21a | 4.42 ± 0.18a | 1.23 ± 0.07b |
| Chaironia | 4.00 ± 0.32b | 2.90 ± 0.28b | 1.50 ± 0.11a |

| Source of variation | | | |
|---|---|---|---|
| Treatment (A) | P < 0.05 | P < 0.05 | Ns |
| Variety (B) | P < 0.05 | P < 0.05 | P < 0.05 |
| AXB (Interaction) | P < 0.05 | P < 0.05 | Ns |

Lower case letters indicate significant differences (p < 0.05); ns: non-significant difference.

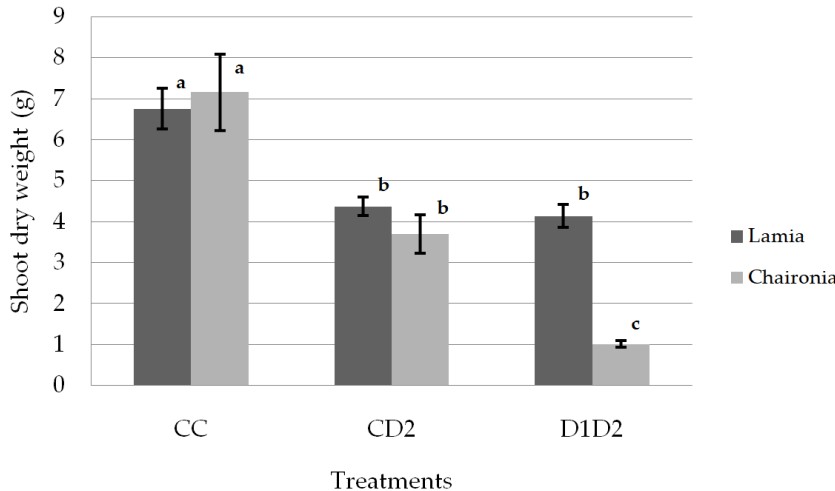

**Figure 2.** Shoot dry weight on day 133. Bars show the mean of the biological replicates ± SE; lower case letters indicate significant differences between means (p < 0.05).

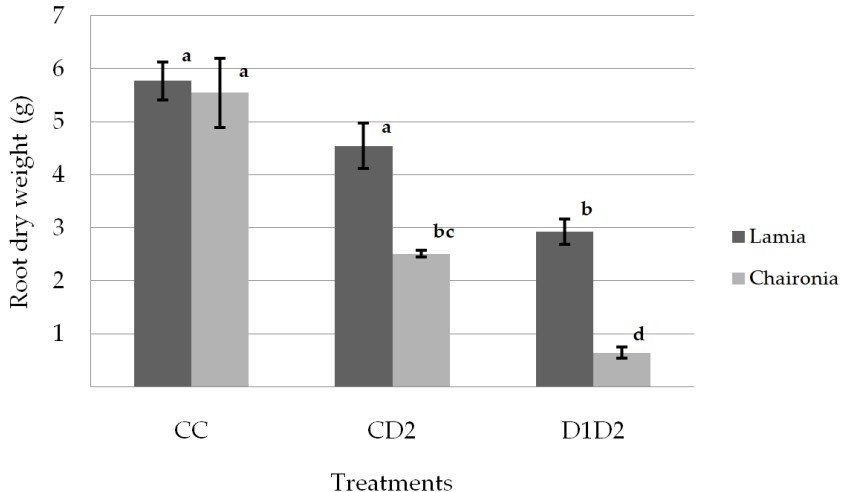

**Figure 3.** Root dry weight on day 133. Bars show the mean of the biological replicates ± SE; lower case letters indicate significant differences between means (p < 0.05).

*3.2. Shoot and Root Water Content*

The Levene test for shoot water content was not significant (p > 0.05) in all cases, indicating homogeneity of the variances. The shoot water content in the CC, CD2, and D1D2 treatments of Lamia seems to be constant, whereas in Chaironia, the CD2 treatment differed significantly from both CC and D1D2 treatments Lamia, displaying remarkably the lowest water content value among all treatments from both varieties. In addition, the shoots of Chaironia D1D2 were better hydrated in comparison with those of Lamia D1D2, and even Lamia CC (Figure 4). On the other hand, there were

no statistically significant differences between all treatments in both varieties, regarding the root water content (Figure 5).

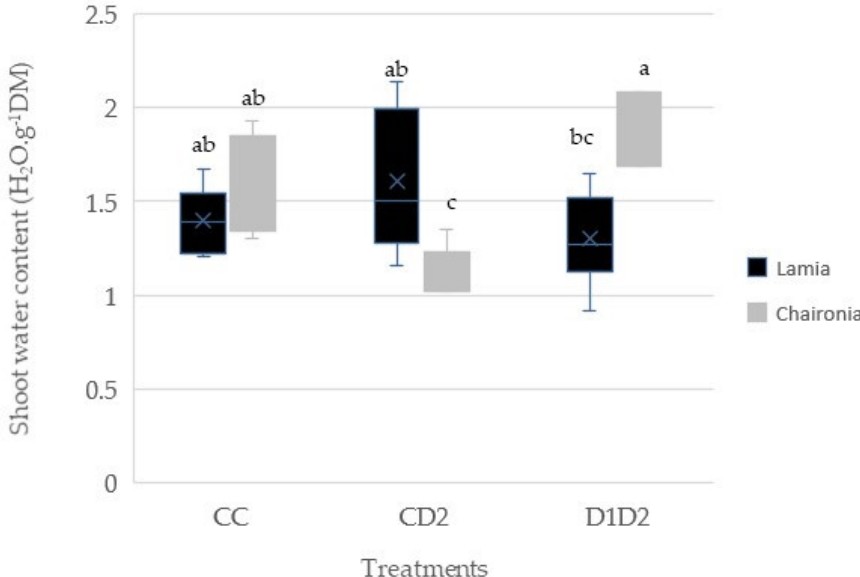

**Figure 4.** Shoot water content expressed as g of water per g of dry shoot weight. Bars show the mean of the biological replicates ± SE on day 133; lower case letters indicate significant differences between the means (p < 0.05).

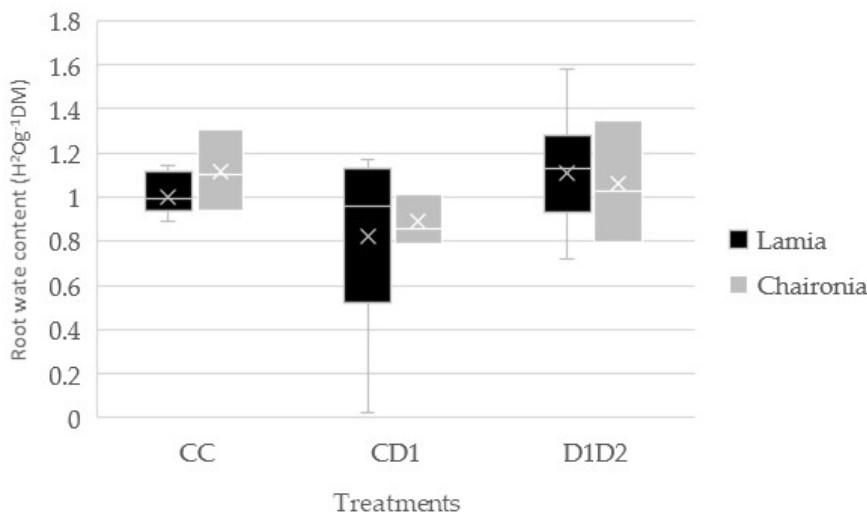

**Figure 5.** Root water content expressed as g of water per g of dry root weight. Bars show the mean of the biological replicates ± SE on day 133. There were no significant differences among the means.

*3.3. MSAP Results*

The two varieties tested (Chaironia and Lamia) presented different patterns of DNA methylation in response to drought stress. In fact, according to the mean value of polymorphic alleles that were provided from the R software with the MSAP_calc program for each category of alleles (h, m and u), Lamia controls presented a 68.57% percentage of total methylation, while a decrease of the total methylation percentage was observed in Lamia D1D2 (61.92%) and CD2 (62.31%). On the other hand,

the total methylation in the leaves of Chaironia controls reached a percentage of 71.12%, whereas a decrease of total methylation was noticed in Chaironia CD2 treatment (58.23%), but not in the case of Chaironia D1D2 (72.36%) (Table 3, Figures 6 and 7).

**Table 3.** DNA methylation polymorphic markers in control (CC) and the two drought treatments (CD2 and D1D2) for the two varieties of *M. sativa.*

| Polymorphic Markers | Lamia CC | Lamia CD2 | Lamia D1D2 | Chaironia CC | Chaironia CD2 | Chaironia D1D2 |
|---|---|---|---|---|---|---|
| **h** methylation | 114 | 87 | 79 | 120 | 79 | 133 |
| **m** methylation | 102 | 80 | 69 | 146 | 66 | 134 |
| Uninformative | 99 | 101 | 91 | 108 | 104 | 102 |
| Total polymorphic markers | 315 | 268 | 239 | 374 | 249 | 369 |
| Total methylation (h + m) | 216 | 167 | 148 | 266 | 145 | 267 |
| Percentage of total methylation % | 68.57 | 62.31 | 61.92 | 71.12 | 58.23 | 72.36 |

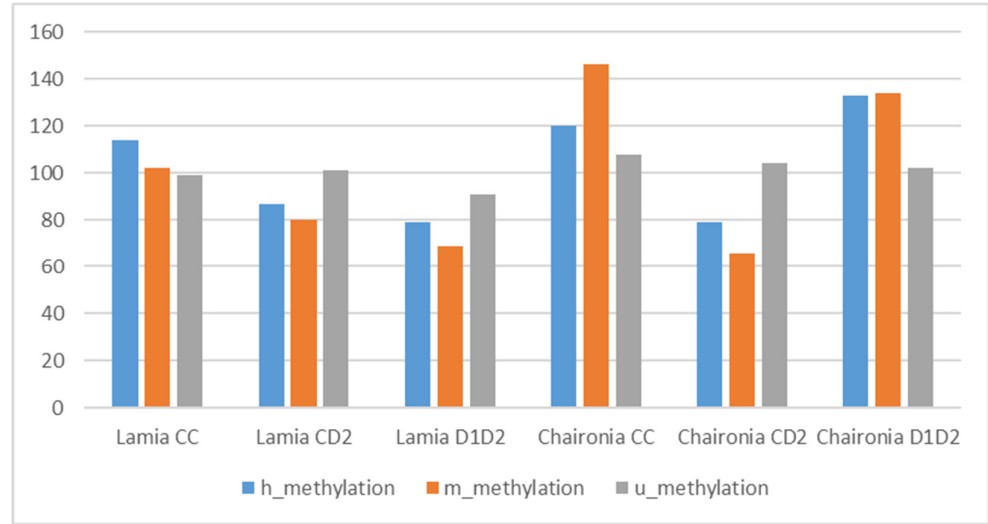

**Figure 6.** Methylation patterns (demonstrated as distinct h/m/u markers) of all treatments tested. Y-axis represents total methylation (h + m alleles).

Furthermore, according to the Kruskal–Wallis test for h alleles for Iepi and hepi, very strong evidence of the difference was provided (p < 0.001) between the mean ranks of at least one pair of groups. For m alleles and total methylation (h + m alleles), the Kruskal–Wallis tests also found significant differences between groups for Iepi and hepi (p < 0.001). Only for uninformative alleles (u alleles), no significant differences were found according to the Kruskal–Wallis test (p = 0.917).

**Table 4.** Mean epigenetic Shannon Information Index (Iepi) and epigenetic diversity (hepi) and standard error (SE) for treatments for each cultivar. The significant differences that are provided from Dunn's post hoc tests with Bonferroni correction among treatments for each variety are indicated with the same letters.

| | | h Alleles | | u Alleles | | m Alleles | | Total Methylation (h + m) | |
|---|---|---|---|---|---|---|---|---|---|
| | | Iepi | hepi | Iepi | hepi | Iepi | hepi | Iepi | hepi |
| **Lamia CC** | **Mean** | 0.168 | 0.109 | 0.194 | 0.127 | 0.168[c] | 0.111[c] | 0.204[f] | 0.142[f] |
| | **SE** | 0.013 | 0.009 | 0.016 | 0.010 | 0.014 | 0.010 | 0.011 | 0.008 |

| | | | | | | | | | |
|---|---|---|---|---|---|---|---|---|---|
| **Lamia CD2** | **Mean** | 0.125 | 0.081 | 0.213 | 0.143 | 0.128 | 0.083 | 0.157 | 0.110 |
| | **SE** | 0.012 | 0.008 | 0.017 | 0.012 | 0.013 | 0.008 | 0.011 | 0.007 |
| **Lamia D1D2** | **Mean** | 0.118 | 0.077 | 0.189 | 0.126 | $0.112^c$ | $0.073^c$ | $0.140^f$ | $0.097^f$ |
| | **SE** | 0.012 | 0.008 | 0.016 | 0.011 | 0.012 | 0.008 | 0.010 | 0.007 |
| **Chaironia CC** | **Mean** | $0.173^a$ | $0.112^a$ | 0.208 | 0.135 | $0.234^d$ | $0.153^d$ | $0.251^g$ | $0.175^g$ |
| | **SE** | 0.013 | 0.009 | 0.016 | 0.010 | 0.015 | 0.010 | 0.012 | 0.008 |
| **Chaironia CD2** | **Mean** | $0.114^{a,b}$ | $0,073^{a,b}$ | 0.212 | 0.140 | $0.105^{d,e}$ | $0.068^{d,e}$ | $0.137^{g,h}$ | $0095^{g,h}$ |
| | **SE** | 0.011 | 0,007 | 0.017 | 0.011 | 0.012 | 0.008 | 0.010 | 0.007 |
| **Chaironia D1D2** | **Mean** | $0.188^b$ | $0.120^b$ | 0.207 | 0.136 | $0.210^e$ | $0.136^e$ | $0.252^h$ | $0.176^h$ |
| | **SE** | 0.013 | 0.008 | 0.016 | 0.011 | 0.014 | 0.009 | 0.012 | 0.008 |

According to Dunn's post hoc tests with Bonferroni correction for Iepi and hepi on each pair of the group, we found that:

a) For h alleles, no significant differences were found between Lamia treatments, while for Chaironia significant differences were found between CC/ CD2, and CD2/D1D2 treatments.

b) For m alleles, significant differences were found for Lamia CC/Lamia D1D2 and for Chaironia CC/Chaironia CD2, Chaironia CD2/Chaironia D1D2.

c) For total methylation (h+m), significant differences were found between Lamia CC/Lamia D1D2, Chaironia CC/Chaironia CD2, Chaironia CD2/Chaironia D1D2.

More specifically for Iepi, Lamia variety exhibited a statistically significant decrease in total (h- and m-alleles) DNA methylation after the second drought episode (D1D2) as compared to the controls (Table 4). In addition, both CD2 and D1D2 treatments showed a decrease in DNA methylation as compared to the initial non-stressed control (CC), although there is no statistically significant difference between total methylation of CD2 and CC. On the other hand, for Iepi and regarding the DNA methylation status of the Chaironia variety, the total methylation (h and m alleles) of CD2 is significantly lower compared to CC, whereas in D1D2 plants, the total methylation has increased significantly as compared to CD2.

*3.4. Among Different Treatments and Between Varieties*

When Dunn's post hoc tests with Bonferroni correction on each pair of groups was employed between different treatments and varieties, significant differences were found for Lamia CC/Chaironia CC, CD2 and D1D2, Lamia CD2/Chaironia CC, and D1D2 and for Lamia D1D2/Chaironia CC and D1D2 (Supplementary Table 1).

Interestingly, from the heat map presentation (Figure 7), three clusters are formed in the variety's domain. Importantly, these clusters present the peculiarity of the Chaironia CD2 to be clustered together with Lamia CC and Lamia CD2 to form a cluster alone.

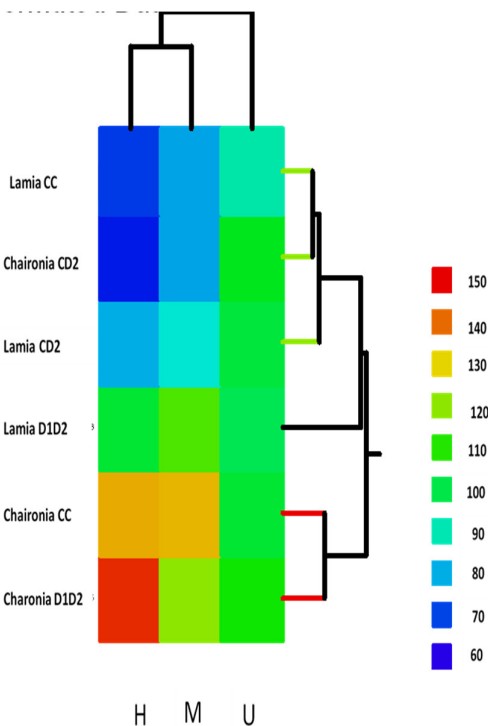

**Figure 7.** The permuted data matrix is actually a heat map presentation of the six different treatments in a three-variable classification space (h-, m-, and u-methylation). The distance measure is the Euclidean distance, and the joining method is the single linkage. Three clusters are formed in the variety's domain. Importantly, these clusters present the peculiarity of the Chaironia CD2 to be clustered together with Lamia CC and Lamia CD2 to form a cluster alone.

## 4. Discussion

In the present study, we investigated the effect of repeated drought stress on two *M. sativa* varieties, at the phenotypic and epigenetic level, in order to determine which is the most drought-tolerant variety and unravel any association of drought tolerance, priming events, and DNA methylation mechanisms.

Plant growth and development is greatly affected by drought stress. Thus, one of the major breeding goals is the improvement of drought stress tolerance. The immediate response to drought stress is relatively well studied, but in nature, drought stress often occurs on a regular basis, and response to this type of stress is much more complicated and less understood. Recent studies have indicated that drought stress priming enhances plant adaptation to subsequent stress by activating different strategies, such as maintaining the photosynthetic capacity [33] accumulating more antioxidants [34] and altering gene expression [35]. The fact that drought priming can improve plant performance to subsequent stress by maintaining its biomass is documented in several studies [36,37].

Drought was imposed by two rounds of reduced water availability (phase I and phase II) with a long recovery phase in between. According to our findings, the Lamia variety seemed to respond better to drought conditions as judged by the comparative evaluation of dry shoot and root weight between the two varieties. Specifically, the Lamia variety displayed significantly higher values of dry shoot and root weight than Chaironia after the two consecutive drought episodes (Figure 2 and 3). Since increased weight is directly linked to higher economic profit for Medicago breeders, the importance of this observation is evident. Moreover, in Lamia, the shoot water content remains constant among the three treatments, whereas Chaironia displayed increased water content in the D1D2 plants that could be attributed to low growth rates and to the overall stunted growth resulting from stress. Collectively, these results suggest that Lamia is probably less affected by drought

treatment compared to Chaironia in terms of compromised yield and quality upon drought, which would make it a preferred variety for breeding purposes.

The results obtained by the MSAP method showed that in the Lamia variety, a significant decrease was observed in DNA methylation upon the second stress incident (D1D2) as compared to the original control (CC) according to Iepi and hepi values. The decrease in DNA methylation might be associated with an attempt to activate specific drought-responsive pathways, as described in previous reports [14,19,38,39,40]. Certainly, further research into identifying the gene sequences underlying these differential DNA methylations and their expression patterns is needed in order to provide evidence for this hypothesis. A decrease was also observed in the D1D2 as compared to CD2, but it was not significant. Overall it may be suggested that the Lamia variety has a trend for DNA methylation decrease when it undergoes repeated drought episodes. Chaironia, on the other hand, undergoes a decline of DNA methylation compared to controls in the first stress episode and then an increase in the second stress, which might reflect an attempt of global transcriptional reprogramming (simultaneous activation and de-activation of genes) rather than maintenance of reduced methylation status observed in the Lamia (Table 4).

In several studies, the decrease of methylation/hypomethylation/demethylation has been proven to be associated with adapted responses to various stresses [41−43]. Similar findings regarding the decrease of methylation percentages, as a probable indication of how drought-resistant plants can cope with stress, are presented for *Lolium perenne* by [12] and for rice genotypes by [44]. Furthermore, in Oryza sativa, it was found that hypomethylation is more frequent in drought-tolerant genotypes [42].

Plants can respond more efficiently to environmental stress by "remembering" a stress episode and react effectively upon subsequent exposures to an environmental challenge There are two types of 'memory'—first, it exists a trans-generational transmission of stress memory to the progeny of a plant that experienced a stressor, and second, during the biological cycle of a plant, there is a short-term memory of a stress episode (known as priming). Such a mechanism allows cells to form a molecular epigenetic memory during the first stress, and the occurrence of second stress activates the stress memory and the induction of a new, improved plant response [8]. Epigenetic marks, such as DNA methylation, can be modified and result in an epigenetic response. Thus, environmental stresses, such as drought, salinity, and elevated temperature, can provoke changes in DNA methylation that lead to modified phenotypes and, ultimately, to better plant adaptation [39,45].

In this work, we attempted to study the priming effect of drought by imposing the two varieties to two consecutive stress episodes allowing a long recovery between the two stress events.

A significant decrease in the dry shoot weight was observed in Lamia upon one drought stress episode (CD2). However, there was no further reduction in the second drought stress. In other words, Lamia is maintaining its 'first stress' biomass in the second drought stress and suffers no further decrease. It is plausible, therefore, that a priming effect may be operating in this system, whereby a drought-responsive mechanism is 'remembered' and activated in the second stress episode and aids in preventing additional biomass decrease. Moreover, the maintenance of decreased total DNA methylation in D1D2 plants may suggest the operation of a DNA methylation 'priming' mechanism in Lamia. This mechanism provides a better and faster adaptation to recurrent stress not by increasing its biomass but by increasing the flexibility of the plant to respond to the environmental stresses (i.e., maintaining shoot and root water content), as indicated in several other studies [39,16].

Although Lamia's adaptation may compromise plant productivity in the short term, it represents increased tolerance to recurrent stresses and, therefore, enhances productivity in the long-term [46,47]. However, whether these drought-induced DNA methylation alterations persist after subsequent recovery, and if they can be transmitted to the next generation needs further investigations. On the other hand, regarding Chaironia variety, a significant decrease is observed in the shoot biomass between CD2 plants and D1D2 plants, according to the Iepi and hepi values. Interestingly, the total methylation is significantly altered (increased) in D1D2 plants compared to CD2. Consequently, the increase in methylation of D1D2 plants compared to CD2 plants in the

Chaironia variety might suggest a negative effect on the total biomass of D1D2 plants. Consequently, no priming effect is observed in the Chaironia variety.

Several reports highlight that extreme environmental constraints, such as drought are expected to occur more frequently and more intensely in the future [48]. Thus, the identification of genetic material that can be adapted to recurrent stressors is of great importance.

Plant adaptation to environmental alterations can be greatly improved through priming, thus reducing economic losses due to stress. However, priming does not always correlate with improved plant performance and, therefore, the memory may only influence the phenotype of specific traits [49].

This study provided baseline knowledge about methylation status, epigenetic memory, shoot and root water content for the two varieties of *M. sativa*, Lamia, and Chaironia. Future research through Whole Genome Bisulfite Sequencing (WGBS) could reveal the differentially methylated regions and underlying genes that are primarily responsible for drought stress giving an important opportunity for future breeding of Medicago species. Moreover, the role of plant hormones (in particular, ABA) [50] in regulating plant adaptation and short stress memory to abiotic stresses through DNA methylation changes should be investigated as well.

**Supplementary Materials:** Table 1. Dunn's post hoc tests with Bonferroni correction for total methylation (h + m alleles) for Iepi

| Total Methylation Significant Differences *Bonferroni Corrected Significance Level: 0.0033* | Chaironia CC | Chaironia CD2 | Chaironia D1D2 |
| --- | --- | --- | --- |
| Lamia CC | **Yes** | **Yes** | **Yes** |
| Lamia CD2 | **Yes** | No | **Yes** |
| Lamia D1D2 | **Yes** | No | **Yes** |

**Author Contributions:** Conceptualization, E. T., E. A. and S. C.; Data curation, Y. V., S. C, G. P. , E.V.A and A. K.; Formal analysis, E.V.A., G.P and E. A.; Funding acquisition, E. T.; Methodology, Y. V., E. T., E.V.A., E.A., S.C and D. V.; Resources, D. V.; Supervision, E. T.; Validation, E. T.; Writing – original draft, all; Writing – review & editing, all    All authors have read and agreed to the published version of the manuscript.

**Funding:** This research received no external funding.

**Acknowledgments:** The authors would like to thank Maro Goufa, Spiridoula Evangelopoulou and Panagioti Koutroumbi for their contribution into greenhouse experiment and data collection.

**Conflicts of Interest:** The authors declare no conflict of interest.

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
