# Peer review of "Recurrent Water Deficit and Epigenetic Memory in Medicago sativa L. Varieties"

_applsci, doi:10.3390/app10093110_

Round 1
Reviewer 1 Report
Minor and major changes are suggested below. In general, the study is interesting, and the work done has merit. Statistics must be clearly improved. The discussion should be more focused on results.
Line 22 add author to species.
Line 30 don't use abbreviations the first time
Line 106-118 it can be included in the plant material section.
Line 138 “were drought treated by withholding water” more details are needed. No water for 60 days? how much water per pot and how often? More detail of the treatments, positive control and drought.
Line 143 Indicate the duration of phase II in days.
Line 150 a clearer explanation of treatments in phase II is necessary. How was soil field capacity measured?.
Line 151 Why was phase II shorter than phase I? How was the duration of the drought period decided? What was the distribution of the plants in the greenhouse, by groups according to treatments or random?
Line 169 “three different plants in each entry” Why only 3 !! being able to choose 7. This reduction of samples decreases the statistical robustness for the following analyzes.
Line 247 Please include more statistical data in order to assess the robustness of the analysis. In table 2 include N (number of data for each mean). Include a normality test on the residuals to see if the Anova analysis can be applied (Shapiro test). If data distribution is not normal, adjust the data to normality by transforming it or use non-parametric methods. Include the Levene test and the eta squared test to see the magnitude of the effect.
Line 251 -158. Probably easier to understand first, talking about differences among treatments within varieties, and then talking about differences between varieties.
Line 266 Means by treatment within variety and their differences are probably more interesting than the global means.
Line 313 In general, barplots do not reveal the distribution of the data. Other data types such as box and whisker charts or violin charts allow a better observation of the distribution of the data.
Line 315 More data is needed to assess the robustness of the analyzes. Indicate the number of data evaluated by treatment and variety (N). Indicate data distribution (normal or not). Carry out Levene, to test the variance homogeneity and see if it is possible to perform out ANOVAS.
Line 315 CD22 = CD2 ?
Line 317 “as well as from “Lamia” CC” When describing the results, mixing differences between treatments with differences between varieties and treatments is confusing.
Line 347 more statistical data is needed to assess the correlation. I do not understand if the correlation is made with only 3 points. In this case, indicate. Provide values for significance (r2, etc.). Is each point the mean of three data? In such a case why not include the 9 points for the calculation of the regression?
Line 352 table 2 = table 3?
Line 394 I do not understand why the results are not expressed as a mean and standard error and differences between treatments. I do not understand why the differences among the means and their significance are not included in the table and not as table footing. The main analysis: differences between the treatments within the variety should be observed in the table. KW analyzes between pairs of treatments are indicated in the table footer, which is quite confusing.
For 2 means comparison, perhaps better to use a test such as Mann-Whitney U test.
Line 380 “while a slight decrease of the total methylation percentage was observed in “Lamia” D1D2 (61.92%) and CD2 (62.31%).” It would be better if we could talk in terms of whether or not there are significant differences among treatments within the variety.
Line 383 the same suggestion as the one made for line 380
Line 396 treatments exposure order has changed with respect to figure 4 and 5
Line 397-402 These analyzes are supposed to be performed on the percentage of total methylation. Indicate somewhere.
Line 398-399 How can it be that Kruskall Wallis test Lamia CC (68) & Lamia CD2 (62.3) is not significant and Kruskall Wallis test Lamia D1D2 (61.9) & Lamia CD2(62.3) is significant?
For me it is not clear whether or not there are significant differences between the treatments
Line 400 Wvalue:9 not significant = Wvalue:not significant
Line 406 Why treatments order has been changed at figure 7? figure 7 can be improve
Line 410 permuted data matrix and cluster analyzes are only multivariate statistical techniques that suggest possible relationships but do not provide significance levels.
Line 455-460. what matters is whether if these small differences are significant or not, if we have studied the effect size, or whether the amount of variation is similar to that of other studies. We are talking about a difference of 61.92 to 62.31 in a sample with 3 data! Is the methodology precise enough to affirm that if we repeat the experiment, the D1D2 treatments will return less methylated than CD2?
Line 458 (Fig 7) Figure 8 is not offering this information
Line 475-478 “Moreover, present results also suggest that chromatin changes (here via DNA methylation) induced by the first drought stress may be stored and maintained in meristematic cells of Medicago stems and leaves and once cutting occurs they are transferred mitotically by cell divisions to the newly formed tissues”
How can this be deduced from the results obtained? Results are not consistent between varieties, or even within varieties. Which results support this statement and how?
Line 480-485 The role of plant hormones is being forgotten.
Line 487 – end. It seems that what is valid for one variety is not valid for the other, in such a way that any result can justify the non-increase, increase or decrease in methylation, all can be justified with previous studies. But what is important is whether the methylation results are consistent and related to those of biomass.
Perhaps trying to answer the following question would be easier and more useful: does the first period of drought prepare the plants to produce more in the second period of drought? that is, when comparing D1D2 with CD2, are the first ones more productive? Is D1D2 more productive than CD2? And does that have a logical correspondence with methylation levels? Were the data on methylation sufficient to establish significant differences?
I think it is better to first focus on the effect of the treatments within varieties, and when that is clearly established then make a comparison between the two varieties, but not mix everything from the beginning
Line 574 incomplete reference
Author Response
Response to Reviewer 1 Comments
Thank you for your constructive comments and suggestions that help to improve our manuscript. We are pleased that the reviewers have positive feedback on manuscript. Please see below for our responses to your comments and suggestions. In the revised manuscript, all changes are distinguished by track changes.
Minor and major changes are suggested below. In general, the study is interesting, and the work done has merit. Statistics must be clearly improved. The discussion should be more focused on results.
Line 22 add author to species.
Response: Done as requested
Line 30 don't use abbreviations the first time
Response: Done as requested
Line 106-118 it can be included in the plant material section.
Response: Done as requested
Line 138 “were drought treated by withholding water” more details are needed. No water for 60 days? how much water per pot and how often? More detail of the treatments, positive control and drought.
Response: Information was added as requested.
Line 143 Indicate the duration of phase II in days.
Response: Done as requested
Line 150 a clearer explanation of treatments in phase II is necessary. How was soil field capacity measured?
Response: Done as requested. Regarding the determination of soil field capacity, information was added in the Materials and Methods section.
Line 151 Why was phase II shorter than phase I? How was the duration of the drought period decided? What was the distribution of the plants in the greenhouse, by groups according to treatments or random?
Response: Explained in the text
Line 169 “three different plants in each entry” Why only 3 !! being able to choose 7. This reduction of samples decreases the statistical robustness for the following analyzes.
Response: According to various published papers like Pan et al. (2011), three replicates are used from each sample to analyze DNA methylation. Moreover, due to financial constraints we analyzed 3 samples.
Pan, Y., Wang, W., Zhao, X., Zhu, L., Fu, B., & Li, Z. (2011). DNA methylation alterations of rice in response to cold stress. Plant Omics J, 4(7), 364-369.
Line 247 Please include more statistical data in order to assess the robustness of the analysis. In table 2 include N (number of data for each mean). Include a normality test on the residuals to see if the Anova analysis can be applied (Shapiro test). If data distribution is not normal, adjust the data to normality by transforming it or use non-parametric methods. Include the Levene test and the eta squared test to see the magnitude of the effect.
Response: Done as requested
Line 251 -158. Probably easier to understand first, talking about differences among treatments within varieties, and then talking about differences between varieties.
Response: Done as requested
Line 266 Means by treatment within variety and their differences are probably more interesting than the global means.
Response: In the table presented the main effects and in the Figures the interactions. Thus, the means by treatment within variety presented in the Figure.
Line 313 In general, barplots do not reveal the distribution of the data. Other data types such as box and whisker charts or violin charts allow a better observation of the distribution of the data.
Response: Done as requested
Line 315 More data is needed to assess the robustness of the analyzes. Indicate the number of data evaluated by treatment and variety (N). Indicate data distribution (normal or not). Carry out Levene, to test the variance homogeneity and see if it is possible to perform out ANOVAS.
Response: Done as requested. Only the root water content was not normally distributed and not parametric test was performed.
Line 315 CD22 = CD2 ?
Response: It is corrected.
Line 317 “as well as from “Lamia” CC” When describing the results, mixing differences between treatments with differences between varieties and treatments is confusing.
Response: Done as requested
Line 347 more statistical data is needed to assess the correlation. I do not understand if the correlation is made with only 3 points. In this case, indicate. Provide values for significance (r2, etc.). Is each point the mean of three data? In such a case why not include the 9 points for the calculation of the regression?
Results were obtained from the program msap_calc program which is written in R, so we do not have the information for each sample in order to insert it in regression analysis but only the mean which is presented here.
Line 352 table 2 = table 3?
Response: Corrected
Line 394 I do not understand why the results are not expressed as a mean and standard error and differences between treatments. I do not understand why the differences among the means and their significance are not included in the table and not as table footing. The main analysis: differences between the treatments within the variety should be observed in the table. KW analyzes between pairs of treatments are indicated in the table footer, which is quite confusing.
Response: Indeed, you have a point here, results are the mean for each treatment for the three plants. Ιn order although to analyze the data we used msap_calc program which is written in R language, the program gives output the mean for the three plants and does not calculate, mean, standard error and differences between treatments. For that reason, we cannot have raw data for polymorphic loci in order to proceed to statistical significance tests. But now we have inserted another table in the manuscript which presents epigenetic diversity (hepi) and epigenetic Shannon Information Index (Iepi). Results were obtained from GeneAlex program for each category of alleles and we proceed to statistical analysis.
For 2 means comparison, perhaps better to use a test such as Mann-Whitney U test.
We used Kruskall- Wallis test in order to examine significant differences between different treatments and cultivars. We further proceed with Dunn’s post hoc tests on each pair of group with Bonferroni correction for h, m alleles and total methylation (h+m alleles) for Iepi.
Line 380 “while a slight decrease of the total methylation percentage was observed in “Lamia” D1D2 (61.92%) and CD2 (62.31%).” It would be better if we could talk in terms of whether or not there are significant differences among treatments within the variety.
Response: Corrected we now discuss about significant differences between treatments of each cultivar according to Iepi and hepi.
Line 383 the same suggestion as the one made for line 380
Response: Corrected we now discuss about significant differences between treatments of each cultivar according to Iepi and hepi.
Line 396 treatments exposure order has changed with respect to figure 4 and 5
Response: Corrected
Line 397-402 These analyzes are supposed to be performed on the percentage of total methylation. Indicate somewhere.
Response: We deleted all those results and we repeated the Kruskall – Wallis test with raw data for hepi and Iepi as it is mentioned above.
Line 398-399 How can it be that Kruskall Wallis test Lamia CC (68) & Lamia CD2 (62.3) is not significant and Kruskall Wallis test Lamia D1D2 (61.9) & Lamia CD2(62.3) is significant?
Response: The results were wrong, we repeated the whole statistical analysis.
For me it is not clear whether or not there are significant differences between the treatments
Response: We added this information in the Table 4 after the new statistical analysis.
Line 400 Wvalue:9 not significant = Wvalue:not significant
Response: The results were wrong, we repeated the whole statistical analysis.
Line 406 Why treatments order has been changed at figure 7? figure 7 can be improve
Response: Corrected
Line 410 permuted data matrix and cluster analyzes are only multivariate statistical techniques that suggest possible relationships but do not provide significance levels.
Response: Indeed, we agree with the reviewer, we present this heatmap only for better visualization of the results.
Line 455-460. what matters is whether if these small differences are significant or not, if we have studied the effect size, or whether the amount of variation is similar to that of other studies. We are talking about a difference of 61.92 to 62.31 in a sample with 3 data! Is the methodology precise enough to affirm that if we repeat the experiment, the D1D2 treatments will return less methylated than CD2?
Indeed, the reviewer has right here, we inserted this information if there are statistical
Response: significant differences and we highlighted the fact that this is a preliminary study with some first results which need further future analysis with other techniques such as bisulfite sequencing etc.
Line 458 (Fig 7) Figure 8 is not offering this information
Response: Corrected
Line 475-478 “Moreover, present results also suggest that chromatin changes (here via DNA methylation) induced by the first drought stress may be stored and maintained in meristematic cells of Medicago stems and leaves and once cutting occurs they are transferred mitotically by cell divisions to the newly formed tissues”
How can this be deduced from the results obtained? Results are not consistent between varieties, or even within varieties. Which results support this statement and how?
Response: We would like to thank the reviewer for his/her valuable comment. Indeed, our intention here was to merely speculate based on the literature that for the Lamia cultivar, changes induced by the first drought episode may be stored in meristematic cells and then after cutting and regrowth perhaps these changes have been maintained and then propagated in the new tissue and enhance the drought response in the second drought episode. Indeed, as the reviewer correctly points out it could not be deduced as we would have to obtain the DNA methylation measurements after the first episode. We have used the estimates for the CD2 plants that only went through one drought occurrence and made a suggestion based on the fact that DNA methylation levels are lower in D1D2 as compared to CD2, for the Lamia cultivar.
Line 480-485 The role of plant hormones is being forgotten.
Response: It is added in the text.
Line 487 – end. It seems that what is valid for one variety is not valid for the other, in such a way that any result can justify the non-increase, increase or decrease in methylation, all can be justified with previous studies. But what is important is whether the methylation results are consistent and related to those of biomass.
Perhaps trying to answer the following question would be easier and more useful: does the first period of drought prepare the plants to produce more in the second period of drought? that is, when comparing D1D2 with CD2, are the first ones more productive? Is D1D2 more productive than CD2? And does that have a logical correspondence with methylation levels? Were the data on methylation sufficient to establish significant differences?
Response: Thank you for your comment. We added a paragraph addressing these comparisons.
I think it is better to first focus on the effect of the treatments within varieties, and when that is clearly established then make a comparison between the two varieties, but not mix everything from the beginning
Line 574 incomplete reference
Response: Corrected
Reviewer 2 Report
The manuscript is interesting and contains a lot of valuable information. The title accurately reflect the content of the article. Abstract fully reflects the essence of the work. The research problem and the purpose of the work have been formulated correctly, the methods adopted are correct and do not raise any objections. The results of the research were presented in a clear manner and correctly interpreted. The study results were confronted correctly and explained with the results of the other authors. The references are properly selected, but there are many mistakes in this section. Other comments and suggestions are in the manuscript.

Author Response
Response to Reviewer 2 Comments
Thank you for your constructive comments and suggestions that help to improve our manuscript. We are pleased that the reviewers have positive feedback on manuscript. Please see below for our responses to your comments and suggestions. In the revised manuscript, all changes are distinguished by track changes.
The manuscript is interesting and contains a lot of valuable information. The title accurately reflect the content of the article. Abstract fully reflects the essence of the work. The research problem and the purpose of the work have been formulated correctly, the methods adopted are correct and do not raise any objections. The results of the research were presented in a clear manner and correctly interpreted. The study results were confronted correctly and explained with the results of the other authors. The references are properly selected, but there are many mistakes in this section. Other comments and suggestions are in the manuscript.
Response: Thank you very much for your comments. All your suggestions/corrections have been incorporated in the text.

Round 2
Reviewer 1 Report
I add some additional comments
Line 138 “were drought treated by withholding water” more details are needed. No water for 60 days? how much water per pot and how often? More detail of the treatments, positive control and drought.
Response: Information was added as requested.
-. Treatments are still not well defined. Especially the drought conditions (water amounts) of phase I.
Line 151 Why was phase II shorter than phase I? How was the duration of the drought period decided? What was the distribution of the plants in the greenhouse, by groups according to treatments or random?
Response: Explained in the text
-. The distribution of the pots has not been commented.
Line 291 Remove Lamia-
Line 315 More data is needed to assess the robustness of the analyzes. Indicate the number of data evaluated by treatment and variety (N). Indicate data distribution (normal or not). Carry out Levene, to test the variance homogeneity and see if it is possible to perform out ANOVAS.
Response: Done as requested. Only the root water content was not normally distributed and not parametric test was performed.
-. number of data evaluated by treatment and variety (N), has not been detailed in any table or place.
Line 347 more statistical data is needed to assess the correlation. I do not understand if the correlation is made with only 3 points. In this case, indicate. Provide values for significance (r2, etc.). Is each point the mean of three data? In such a case why not include the 9 points for the calculation of the regression?
Results were obtained from the program msap_calc program which is written in R, so we do not have the information for each sample in order to insert it in regression analysis but only the mean which is presented here
-. Not a very convincing answer. A correlation without its statistical parameters has little value. You should be able to control the statistical analyzes that are performing.
-. This section should go after this other section 3.4 MSAP Results.
Line 394 I do not understand why the results are not expressed as a mean and standard error and differences between treatments. I do not understand why the differences among the means and their significance are not included in the table and not as table footing. The main analysis: differences between the treatments within the variety should be observed in the table. KW analyzes between pairs of treatments are indicated in the table footer, which is quite confusing.
Response: Indeed, you have a point here, results are the mean for each treatment for the three plants. Ιn order although to analyze the data we used msap_calc program which is written in R language, the program gives output the mean for the three plants and does not calculate, mean, standard error and differences between treatments. For that reason, we cannot have raw data for polymorphic loci in order to proceed to statistical significance tests. But now we have inserted another table in the manuscript which presents epigenetic diversity (hepi) and epigenetic Shannon Information Index (Iepi). Results were obtained from GeneAlex program for each category of alleles and we proceed to statistical analysis.
-. Not a very convincing answer. Therefore, if we cannot talk of significant differences in Table 3, the results and discussion are limited. Again, the “program used” is limiting our knowledge and control on the statistical analysis.
Response: Corrected we now discuss about significant differences between treatments of each cultivar according to Iepi and hepi.
-. The description of the Iepi and hepi differences is quite confusing. Table 4 is confusing as the letters of significance do not know what they refer to (rows, columns, pairs, total?)
-. Line 404 , = and
-. Line 427 a point is missing.
Line 436-439
The results obtained by the MSAP method suggest that the Lamia and Chaironia genotypes respond to drought stress in an epigenetic manner which involves the mechanism of DNA methylation that may be associated to activation of drought response processeso as to better cope with the imposed stresses
-. Long sentence, difficult to understand, generic, is not based on concrete results. It predisposes the discussion without looking at the detail of the results.
-. In general, the discussion is very speculative, and the most important thing, little or nothing based on whether the differences are significant or not and not taking into account the correlation between biomass and methylation.
-. The most important thing is that we cannot support the hypothesis:
The results obtained by the MSAP method suggest that the Lamia and Chaironia genotypes respond to drought stress in an epigenetic manner
Because:
-.the differences among treatments are very small,
-.the results are not clear or correlated (MSAP-Biomass)
-. each variety behaves different
-. In Lamia there seems to be a MSAP pattern (although differences are minimal), however there are no significant differences in biomass. Is this supposed “priming methylation” preparing the plant for the second Fase II period?
-. L489 Chaironia variety, a significant decrease is observed in shoot biomass between CD2 plants and D1D2 plants
Thus, in one variety (Lamia) we don't see an effect from CD2 to D1D2 and in the other Chaironia the biomass decreases, where is the "priming" effect? How have plants adapted after the first dry period? Has the first dry period served to improve the response to the second dry period?
Answering this last question is very important. Seeing the biomass results, I would say no. That the first dry fase I have not served to improve the response to the second dry period. Therefore, what is the point of methylation? Supposed methylation was useful for the plants?
This should be discussed in the discussion.
Perhaps in this experiment, the result is just the opposite of the expected one: in this case, a clear response and adaptation of the plants to the drought period is not observed, nor in the biomass parameters, not in the methylation levels (since the differences they are very small and inconsistent).

Author Response
Line 138 “were drought treated by withholding water” more details are needed. No water for 60 days? how much water per pot and how often? More detail of the treatments, positive control and drought.
Response: Information was added as requested.
Treatments are still not well defined. Especially the drought conditions (water amounts) of phase I.
Response 2: Thank you for your comment. We added some information regarding the drought conditions of phase I.
Line 151 Why was phase II shorter than phase I? How was the duration of the drought period decided? What was the distribution of the plants in the greenhouse, by groups according to treatments or random?
Response: Explained in the text
The distribution of the pots has not been commented.
Response 2: Added in the text. The pots were placed completely randomized
Line 291 Remove Lamia-
Response 2: Removed.
Line 315 More data is needed to assess the robustness of the analyzes. Indicate the number of data evaluated by treatment and variety (N). Indicate data distribution (normal or not). Carry out Levene, to test the variance homogeneity and see if it is possible to perform out ANOVAS.
Response: Done as requested. Only the root water content was not normally distributed and not parametric test was performed.
number of data evaluated by treatment and variety (N), has not been detailed in any table or place.
Response 2: Thank you for the comment. It is now added to the text. Seven replicates for each treatment for each variety.
Line 347 more statistical data is needed to assess the correlation. I do not understand if the correlation is made with only 3 points. In this case, indicate. Provide values for significance (r2, etc.). Is each point the mean of three data? In such a case why not include the 9 points for the calculation of the regression?
Results were obtained from the program msap_calc program which is written in R, so we do not have the information for each sample in order to insert it in regression analysis but only the mean which is presented here
-. Not a very convincing answer. A correlation without its statistical parameters has little value. You should be able to control the statistical analyzes that are performing.
-. This section should go after this other section 3.4 MSAP Results.
Response 2: Thank you for the comment. We removed all this section, as it was not possible to use the 9 points in the analysis.
Line 394 I do not understand why the results are not expressed as a mean and standard error and differences between treatments. I do not understand why the differences among the means and their significance are not included in the table and not as table footing. The main analysis: differences between the treatments within the variety should be observed in the table. KW analyzes between pairs of treatments are indicated in the table footer, which is quite confusing.
Response: Indeed, you have a point here, results are the mean for each treatment for the three plants. Ιn order although to analyze the data we used msap_calc program which is written in R language, the program gives output the mean for the three plants and does not calculate, mean, standard error and differences between treatments. For that reason, we cannot have raw data for polymorphic loci in order to proceed to statistical significance tests. But now we have inserted another table in the manuscript which presents epigenetic diversity (hepi) and epigenetic Shannon Information Index (Iepi). Results were obtained from GeneAlex program for each category of alleles and we proceed to statistical analysis.
Response 2: We take into consideration the remark but we would like to retain Table 3 in the manuscript although it cannot be supported with statistical inference. In our point of view descriptive statistics present valuable information for methylation profiles that shouldn’t get excluded from the manuscript (see e.g. Tang et al (2014), Wang et al (2015), Neves et al (2017))
Furthermore, we inserted parameters like epigenetic Shannon diversity index and epigenetic diversity (hepi) which were calculated with GeneAlex. GeneAlex is widely used for genetic and epigenetic parameters calculations and for example another publication from Zang et al (2015) used GeneAlex in order to proceed with Principal Coordinate Analysis and they also did not perform any statistical test for MSAP.
Recently there is also a strong opinion and trend from worldwide scientists that supports the value of descriptive statistics presentation (numerical summaries of data etc.) (see e.g. http://amstat.tandfonline.com/doi/abs/10.1080/00031305.2016.1154108#.Vt2XIOaE2MN)
Neves, D.M., Almeida, L.A.d.H., Santana-Vieira, D.D.S. et al. Recurrent water deficit causes epigenetic and hormonal changes in citrus plants. Scientific Reports 7, 13684 (2017). https://doi.org/10.1038/s41598-017-14161-x
Wang, W., Huang, F., Qin, Q., Zhao X., Li Z., Fu B. Comparative analysis of DNA methylation changes in two rice genotypes under salt stress and subsequent recovery. Biochemical and Biophysical Research Communications,465, 790-796 (2015).
Tang, X., Tao, X., Wang, Y. et al. Analysis of DNA methylation of perennial ryegrass under drought using the methylation-sensitive amplification polymorphism (MSAP) technique. Mol Genet Genomics 289, 1075–1084 (2014). https://doi.org/10.1007/s00438-014-0869-6
Zhang P.Y., Wang J.G., Geng Y.P., et. al. MSAP-based analysis of DNA methylation diversity in tobacco exposed to different environments and at different development phases. Biochemical Systematics and Ecology 62,249-260 (2015).
-. Not a very convincing answer. Therefore, if we cannot talk of significant differences in Table 3, the results and discussion are limited. Again, the “program used” is limiting our knowledge and control on the statistical analysis.
Response 2: Corrected we now discuss about significant differences between treatments of each cultivar according to Iepi and hepi.
-. The description of the Iepi and hepi differences is quite confusing. Table 4 is confusing as the letters of significance do not know what they refer to (rows, columns, pairs, total?)
Response 2: We further inserted information about Iepi and we have a text which point out the differences for Iepi and alleles (h, m& h+m) below the Table 4. The letters of significance in Table 4 indicate significant differences across columns. In Supplementary Table 1 we have differences among different treatments and between varieties and we decided to show these differences in supplementary section in order not to put so many letters to highlight the statistical significant differences and confuse the readers.
-. Line 404, = and
Response 2: corrected
-. Line 427 a point is missing.
Response 2: corrected
Line 436-439
The results obtained by the MSAP method suggest that the Lamia and Chaironia genotypes respond to drought stress in an epigenetic manner which involves the mechanism of DNA methylation that may be associated to activation of drought response processes as to better cope with the imposed stresses
-. Long sentence, difficult to understand, generic, is not based on concrete results. It predisposes the discussion without looking at the detail of the results.
Response 2: We thank the editor for his/her valuable comment. Indeed, we have now changed the sentence in order to focus more on results (new sentence in text).
-. In general, the discussion is very speculative, and the most important thing, little or nothing based on whether the differences are significant or not and not taking into account the correlation between biomass and methylation.
Response 2: Thank you for the comment, we have removed a lot of discussion parts that were a bit speculative and we have focused on significant differences
.
-. The most important thing is that we cannot support the hypothesis:
The results obtained by the MSAP method suggest that the Lamia and Chaironia genotypes respond to drought stress in an epigenetic manner
Response 2: Indeed, as mentioned before we have changed this sentence, nevertheless we would like to address and clarify the specific points made by the reviewer as they are shown below:
Because:
-.the differences among treatments are very small,
-. For Lamia:
With regard to shoot dry weight and drought responsiveness:
For the Lamia cultivar and with respect to the shoot dry weight, there is a significant decrease between the original control CC and CD2 (one stress episode) and between the original control CC and D1D2 (two stress episodes). However, there are no significant differences between CD2 and D1D2. That is, Lamia does not suffer further reduction in dry weight upon the second stress treatment. It is possible therefore that Lamia responds to recurrent stress drought in an effective way since it does not loose biomass in a second stress.
With regard to DNA methylation and drought responsiveness:
For the Lamia cultivar and with respect to DNA methylation as determined by MSAP analysis, there is a significant decrease between the original control CC and the second stress (D1D2).
So it is possible that a response to the second drought stress and the fact that there is no further decrease in biomass in the second stress may be associated with a decrease in DNA methylation. Thus we could even suggest that Lamia upon a second stress responds in an epigenetic manner since DNA methylation is an epigenetic mechanism.
The relation between drought-tolerant responsiveness and DNA methylation decrease has been reported in other studies which are mentioned in the text and are also given below.
References:
Choi, C.-S.; Sano, H. Abiotic-stress induces demethylation and transcriptional activation of a gene encoding a glycerophosphodiesterase-like protein in tobacco plants. Molecular Genetics and Genomics 2007, 277, 589-600, doi:10.1007/s00438-007-0209-1.
Garg, R.; Narayana Chevala, V.V.S.; Shankar, R.; Jain, M. Divergent DNA methylation patterns associated with gene expression in rice cultivars with contrasting drought and salinity stress response. Scientific Reports 2015, 5, 14922, doi:10.1038/srep14922.
Uthup, T.K.; Ravindran, M.; Bini, K.; Thakurdas, S. Divergent DNA Methylation Patterns Associated with Abiotic Stress in Hevea brasiliensis. Molecular Plant 2011, 4, 996-1013, doi:10.1093/mp/ssr039.
With regard to DNA methylation and priming effects:
However in Lamia, there is a decrease but not significant between original CC plants and first drought stress (CD2) and between CD2 and second stress (D1D2). So we can only speculate that there might be a gradual decrease in DNA methylation during the first stress ‘remembered’ in the second stress.
For Chaironia:
For the Chaironia cultivar and with respect to shoot dry weight: a significant decrease in plant biomass between the first stress (CD2) and the second stress (D1D2) is observed. Regarding DNA methylation, a significant increase in total DNA methylation is observed in D1D2 plants compared to CD2 plants and almost near to original control CC plants. Thus, no correlation between shoot biomass and DNA methylation can be made at present.
-the differences among treatments are very small,
The differences among treatments give an indication of decrease of total DNA methylation in Lamia variety upon recurrent stress
There is a tendency of decrease of total methylation in Lamia variety
-.the results are not clear or correlated (MSAP-Biomass)
For the Lamia cultivar there is a significant decrease in DNA methylation in the second stress as compared to original control.
There is also a maintenance in biomass (no further decrease in biomass, stays at ‘first stress biomass’) upon the second stress.
It is plausible then, that in Lamia, DNA methylation may have activated mechanisms that provide drought tolerance and prevent biomass reduction upon the second stress.
-. each variety behaves different
Response 2: It is a very good point. Indeed, each variety behaves differently. So, it seems the response to drought stress is genotype-specific and this is now added in the text. Regarding Chaironia there is a significant DNA methylation decrease in CD2 as compared to CC and a significant increase in D1D2 as compared to CD2 which is a different pattern than that observed in Lamia. This may have a relation to the fact that Chaironia suffers an even further decrease in biomass upon second stress, which is contrary to what happens in Lamia. Thus, the different DNA methylation levels in Chaironia may be associated with the different response to stress in this genotype as compared to Lamia. DNA methylation alterations upon stress are known to depend on a variety of parameters such as cell-type, tissue, developmental stage as well as different genotype.
-. In Lamia there seems to be a MSAP pattern (although differences are minimal), however there are no significant differences in biomass. Is this supposed “priming methylation” preparing the plant for the second Phase II period.
Response 2: As mentioned above in Lamia, there are significant decreases in biomass between CC and CD2, and between CC and D1D2. There are not significant differences between CD2 and D1D2 which only indicates that biomass is maintained and not reduced upon a second stress and therefore suggests a more tolerant drought response for Lamia upon a second stress.
In terms of MSAP analysis, in Lamia a significant decrease is observed in D1D2 plants as compared to CC. This suggest that 1) there is an association between DNA methylation status and drought response 2) in conjunction with the biomass data it suggests that decrease in DNA methylation may have activated drought-responsive protective mechanisms in second stress that prevent biomass reduction.
In addition, as mentioned above in Lamia, significant decrease is observed in biomass between the original control (CC) and the first stress (CD2) and between original control and second stress (D1D2). There are no differences between the first stress (CD2) and second stress episode (D1D2) suggesting that there might be a priming effect that is remembered and activated in the second stress which contributes to drought tolerance and maintain biomass and prevent further reduction.
Also in terms of DNA methylation there is a significant decrease upon second stress (D1D2) as compared to the original control (CC). There is a decrease but not significant between first stress (CD2) and second stress (D1D1). So we can only speculate that DNA methylation (priming methylation) is involved in the mechanisms underlying a priming effect.
It is now clearly mentioned in the text that epigenetic priming (demonstrated in Lamia as DNA methylation decrease) is improving its adaptation to recurrent drought stresses. Thus it may compromise plant productivity in the short term, but it gives increased tolerance to recurrent stresses and therefore enhances productivity in the long term as mentioned in several other cases. In none of the works cited, it is mentioned about increased productivity of the plants having ‘memory’, they only talk about better plant adaptation.
-. L489 Chaironia variety, a significant decrease is observed in shoot biomass between CD2 plants and D1D2 plants
Response 2: Indeed, this is so. And this is what differentiates it from cultivar Lamia. As mentioned above DNA pattern is also different. The fact that DNA methylation is not decreased in D1D2 may imply that in this case drought responsive mechanisms to protect biomass decrease in Chaironia cannot be activated.
Thus, in one variety (Lamia) we don't see an effect from CD2 to D1D2 and in the other Chaironia the biomass decreases, where is the "priming" effect? How have plants adapted after the first dry period? Has the first dry period served to improve the response to the second dry period?
Response:
We thank the reviewer for this comment because it gives us the opportunity to explain further the results.
Yes, it improved Lamia’s adaptability to the recurrent stress by maintaining its ‘first stress’ biomass in the second drought stress and suffers no further decrease. It is plausible therefore that a priming effect may be operating in this system whereby a drought-responsive mechanism is ‘remembered’ and activated in the second stress episode and aids in preventing additional biomass decrease. Curiously, DNA methylation levels were decreased significantly in the second drought stress. This may suggest the operation of a DNA methylation ‘priming’ mechanism in Lamia.
On the other hand Chaironia behaves in a different way than Lamia. Chaironia may not be able to activate priming mechanisms and beneficial drought-responsive mechanisms under these conditions or to the extent that Lamia does. However, this does not mean that they do not operate in other genotypes like Lamia.
So why the decrease in DNA methylation in Chaironia in the first stress? Probably DNA methylation alterations in Chaironia affect other gene-networks than Lamia, or inherent genetic differences in Chaironia may not allow DNA methylation reduction to lead to the same effects as in Lamia or they partly allow it but there are other pathways affected as well. Further research is needed to identify those gene-networks and metabolic pathways specific for each genotype that are affected by DNA methylation alterations upon drought stress in order to answer explicitly this question.
Answering this last question is very important. Seeing the biomass results, I would say no. That the first dry phase I have not served to improve the response to the second dry period. Therefore, what is the point of methylation? Supposed methylation was useful for the plants
The first dry phase has improved Lamia’s response to the second dry period as it maintained its ‘first stress’ biomass in the second drought stress and suffers no further decrease. The decrease of DNA methylation may have activated the mechanisms required to prevent further biomass decrease. Chaironia presents with a different pattern and it is as if the first stress has not served to improve response to second stress.
Therefore what is the point of methylation?
Answered in previous comment response. Also, global changes in DNA methylation can activate or silence a multitude of genes. Sometimes it is useful and sometimes it is not depending on which genes are activated or silenced, under which conditions and spatio-temporal parameters and which genotypes. For example decrease of DNA methylation at transposable elements (TE) sequences and subsequent TE activation maybe detrimental for an organism. Based on previous research on plants (mentioned in previous comment response) drought responsive decrease of DNA methylation was a beneficial response. However, these responses also depend on individual genotypes.
This should be discussed in the discussion.
Response: It is answered in previous response.
Perhaps in this experiment, the result is just the opposite of the expected one: in this case, a clear response and adaptation of the plants to the drought period is not observed, nor in the biomass parameters, not in the methylation levels (since the differences they are very small and inconsistent).
